# Can social support buffer the association between loneliness and hypertension? a cross-sectional study in rural China

Aki Yazawa[1,2]*, Yosuke Inoue[1,3], Taro Yamamoto[4], Chiho Watanabe[1,5], Raoping Tu[4,6], Ichiro Kawachi[2]

1 Department of Human Ecology, Graduate School of Medicine, The University of Tokyo, Bunkyo-ku, Tokyo, Japan, 2 Department of Social and Behavioral Sciences, Harvard T. H. Chan School of Public Health, Boston, MA, United States of America, 3 Department of Epidemiology and Prevention, Center for Clinical Sciences, National Center for Global Health and Medicine, Shinjuku-ku, Tokyo, Japan, 4 Department of International Health, Institute of Tropical Medicine (NEKKEN), Nagasaki University, Nagasaki-shi, Nagasaki, Japan, 5 National Institute for Environmental Studies, Tsukuba-shi, Ibaraki, Japan, 6 School of Nursing, Yangzhou University, Yangzhou, China

* aki.yazawa@gmail.com

**Data Availability Statement:** The dataset has ethical and legal restrictions for public deposition due to the inclusion of sensitive information from the human participants. These restrictions are

## Abstract

### Objectives

Hypertension has reached epidemic levels in rural China, where loneliness has been a major problem among community dwellers as a consequence of rural-to-urban migration among younger generations. The objective of the study is to investigate the association between loneliness and hypertension, and whether social support can buffer the association (i.e., stress buffering theory), using cross-sectional data from 765 adults (mean age: 59.1 years) in rural Fujian, China.

### Methods

Social support was measured as the reciprocal instrumental social support from/to neighbors and the reciprocal emotional support (i.e., the number of close friends that the respondent could turn to for help immediately when they are in trouble). A mixed-effect Poisson regression model with a robust variance estimator was used to investigate the association between loneliness, social support, and hypertension.

### Results

Analysis revealed that those who were lonely had a higher prevalence ratio for hypertension (prevalence ratio = 1.12, 95% confidence interval 0.99–1.26) compared to those who reported not being lonely. There was an interaction between social support and loneliness in relation to hypertension. Specifically, contrary to the stress buffering theory, the positive association between loneliness and hypertension was more pronounced among those who reported higher social support compared to those who reported lower support (*p* for interaction <0.001 for instrumental support).

imposed by local health authorities in China. Data are available from the first author (AY) upon reasonable request. An inquiry regarding the dataset can also be sent to the Ethics Committee for Medical Research at the University of Tokyo (ethics@m.u-tokyo.ac.jp) who approved this study.

**Funding:** This study was financially supported by the JSPS KAKENHI from the Japan Society for the Promotion of Science (JSPS KAKENHI Grant Number JP13J06172; recipient: AY), and National Center for Global Health and Medicine (Grant Number 21A1020; recipient: YI). The funders had no role in study design, data collection and analysis, decision to publish, or preparation of the manuscript. AY is financially supported by Japan Society for the Promotion of Science as a JSPS Overseas Research Fellow.

**Competing interests:** The authors have declared that no competing interests exist.

## Conclusion

The results suggest that being lonely despite high levels of social support poses the greatest risk for hypertension. This study did not confirm a buffering effect of social support on the association between loneliness and hypertension.

## Introduction

More than 1.13 billion people are affected with hypertension globally [1]. Among risk factors for hypertension (e.g., salt in the diet, overweight/obesity, tobacco use, and physical inactivity) [2], recent studies have suggested that loneliness, which has been defined as the subjective feeling that accompanies the perception that one's social needs are not being met by the quantity or quality of one's social relationships [3], may be an important risk factor for hypertension [4–6]. Chronic loneliness is considered as a psychological stressor, which could lead to worse health through behavioral choices (e.g., less physical activity, more daily smoking, and poor sleep), and could also be directly linked to impaired stress response (i.e., physiological functioning) [3]. Meta-analyses have shown that loneliness is a risk factor for coronary heart disease, stroke [7] and all-cause mortality [8].

In China, loneliness has been singled out as a major problem among rural community dwellers [9]. The combination of China's one-child policy (from the years 1979 to 2015) and the massive rural-to-urban migration of working-aged adults has resulted in a high prevalence of socially isolated 'left behind' people, especially among older population [10]. According to surveys, the prevalence of loneliness among older people in rural areas ranges from 25% to 78% [9, 11]. These sociodemographic transitions might have underlain the increase in disease burden associated with hypertension; mortality attributable to hypertension in China almost doubled during the decade from 2007 to 2017 [12], with a larger increase observed in rural vs. urban areas [13, 14]. Despite this, to date, there have been no studies that examined the association between loneliness and hypertension in China.

In the absence of family support especially from adult children both mentally and instrumentally, social support in the community (i.e., the resources provided by one's network with the intention of increasing one's coping ability [15]) can be a key to mitigate the adverse effect of loneliness in rural Chinese society. Social support, which is commonly categorized into several types of behaviors (e.g., instrumental support and emotional support [16]), is suggested to be a coping resource to overcome loneliness [17] and affect health via a buffering mechanism [18]. Although the studies are not uniformly positive, some studies support a buffering effect of social support on the association between perceived stress and ambulatory blood pressure [19], as well as low income and diastolic blood pressure [20].

Against this background, the purpose of this study was to (1) investigate the association between loneliness and hypertension in rural Fujian, China, and to (2) investigate the potential buffering effect of social support in the association between loneliness and hypertension. The authors hypothesized that (1) loneliness is positively associated with hypertension; and (2) social support from neighbors would mitigate (buffer) the association between loneliness and hypertension (i.e., high social support would protect against the positive association of loneliness on hypertension).

## Materials and methods

### Field survey

A cross-sectional survey was conducted in a sample of seven rural communities in one city in Fujian Province, China in August 2015, selected on the basis of average population size and

level of economic development. Median household income based on the responses from 161 household heads (i.e., 11,000 RMB, interquartile range: 4800–24000 RMB) was comparable to figure reported for rural communities in the area as of 2014 (i.e., 11,252 RMB) (1 RMB = 0.16 USD as of 2015) [21]. All residents aged 18 years or older (i.e., adults in China) were invited through an advertisement posted at each community health center, and about 62% of the population eventually participated (i.e., the convenience sampling). Questionnaire and anthropometric data were collected by trained staff from 797 participants. Details of the survey were described elsewhere [22].

## Measures

**Hypertension.** Systolic and diastolic blood pressure (SBP and DBP) were measured in the sitting position with the left arm held horizontal at the level of the heart using an automated oscillometric monitor by the authors or staffs at local health centers (HEM-7000, OMRON Corp., Japan). They were measured twice and the mean values were calculated. Although there was no specific resting time before the measurement, people first registered and took instructions/explanations about the survey and then signed the certificate of consent before the measurement (~10 min/person). People were defined as having hypertension if they had SBP ≥140 mmHg or DBP ≥90 mmHg or if they were currently taking antihypertensive medication.

**Loneliness.** Loneliness was measured using the single item: 'How often did you feel lonely in the past one month?' with possible responses on a 5-point Likert scale. The answers were dichotomized: lonely (frequently and always, or sometimes) vs. not lonely (never, rarely). The single-item self-report measure has been shown to be highly correlated with an established loneliness scale (the University of California Los Angeles (UCLA) Loneliness Scale) comprising 20 items (e.g., r = 0.72) [23], and is widely used in field surveys [24].

**Social support.** Social support was measured in two dimensions: reciprocal instrumental support and reciprocal emotional support. Reciprocal instrumental support was measured as the reported frequency of exchanging (receiving or providing) various commodities (e.g., food and medicines) between neighbors. Data on the frequency of receiving and providing instrumental support were separately obtained to create four categories (i.e., neither receiving or providing instrumental support; only receiving instrumental support; only providing instrumental support; both receiving and providing instrumental support). Reciprocal emotional support was assessed as the number of close friends that the respondent could turn to for help immediately when they are in trouble, which was then categorized into three groups (none; 1 to 5; 6 or more) since the number of friends ranged from 0–35, and 48% of the participants answered they have no friend.

**Demographic, socio-economic, and lifestyle factors.** Demographic and socio-economic data included age (in years), age-squared, sex (male; female), body mass index (kg/m$^2$) which was categorized into four levels according to Asian cut-off values (underweight: <18.5; normal weight: 18.5–22.9; overweight: 23.0–27.5; obese: ≥27.5) [25], marital status (has a partner; not married; divorced or widowed), educational attainment (illiterate; less than elementary school; junior high school or higher), employment status (not currently employed; farming/fishing; self-employed; formal employee; part-time job with heavy physical activity (e.g., construction workers), part-time job with low-moderate physical activity (e.g., office workers); others), and household income, which was self-reported on a 10-point Likert scale and categorized into tertiles (low; middle; high). Lifestyle factors included alcohol consumption (does not drink; 1 or 2 days a week; 3 to 6 days a week; every day (<50g pure alcohol); every day (≥50g pure alcohol)), smoking (never smoked; stopped smoking; currently smoke), and physical activity. The

amount of alcohol was calculated if the participant answered that they consume alcoholic beverages every day. For the usual daily quantity consumed, participants were asked to report types (beer (~4%), rice wine (~25%), strong spirits (~50%), wine (~10%)) and quantity (bottle for beer, which is usually 640 ml in China, and liang (Chinese ounce equivalent with 50g) or for others). Pure alcohol consumption was then calculated and those who consumed 50g or more pure alcohol daily were defined as having heavy drinking habit [26]. Physical activity was measured by a question 'Compared to other people in your village, how do you rate your own physical activity level?' on a 10-point Likert scale, and then categorized into three groups (0–3; 3.5–5.5, 6–10) to roughly categorize them into inactive/normal/relatively active groups.

## Statistical analysis

After excluding people with missing values on loneliness, hypertension, social support, and covariates (n = 34), the sample size for the analysis was 763. Those who were excluded were more likely to be older, have lower education, have no partner, have lower physical activity, and to be underweight. Infirmity and difficulties in communication were the main reasons for missing data.

A mixed-effect Poisson regression analysis with a random effect and robust variance estimator [27] was used to investigate the association between loneliness, social support and hypertension, since the prevalence of hypertension was high (46%) among the participants [28]. A random effect model was chosen, given the large variations in the levels of urbanization across communities (the proportion of those engaging in farming/fishing: 53.2–80%) and population size (500–1220). Covariates included age, age-squared, sex, body mass index, marital status, educational attainment, employment, household income, alcohol consumption, smoking, and physical activity. Relative excess risk due to interaction (RERI) was also calculated using estimates from the Poisson regression to investigate whether additive interaction is positive or negative [29, 30].

Model 1 analyzed the association between loneliness and hypertension. Models 2 and 3 then included the interaction terms between loneliness and instrumental support and emotional support, respectively. We also conducted sensitivity analyses. First, a least-squares linear regression with a random effect was used to investigate the association between loneliness and hypertension by using log-transformed SBP and DBP as outcomes. Second, covariates were excluded to see the uncontrolled association between main variables. Third, different cut-points were used for loneliness (i.e., frequently and always vs. sometimes, rarely, and never) to see if the findings were robust. Fourth, the age-stratified analysis (younger than 45 years old, 45 to 64 years, and 65 years or older) were conducted. Finally, the analysis with community fixed effects rather than random effects was conducted.

All statistical analyses were conducted using Stata 16.1 (StataCorp, College Station, TX, USA). The level of statistical significance was set at $p < 0.05$ (two-tailed).

## Ethical approval

All procedures performed in studies involving human participants were in accordance with the ethical standards of the institutional and/or national research committee and with the 1964 Helsinki declaration and its later amendments or comparable ethical standards. The protocol was reviewed and approved by the human subjects committees of the Chinese government, Ethics Committee for Medical Research at the University of Tokyo (No. 10515-(1)) and the Ethics Committee of the Institute of Tropical Medicine at Nagasaki University (No. 120910100). Written informed consent was obtained from all individual participants included in the study.

## Results

### Characteristics of study participants

Table 1 summarizes the characteristics of the study participants. The mean age was 59.0 years, and males comprised 38.9% of the sample. Thirty-nine percent were categorized as lonely,

**Table 1. Basic characteristics of the study participants (n = 763).**

|  | Total | Loneliness | |
|---|---|---|---|
|  | (n = 763) | Yes (n = 295) | No (n = 468) |
| Age (in years) | 59.0 [12.9] | 61.7 [12.0] | 57.4 [13.2] |
| Sex (Male) | 297 (38.9) | 106 (35.9) | 191 (40.8) |
| BMI category |  |  |  |
| Underweight | 35 (4.6) | 12 (4.1) | 23 (4.9) |
| Normal | 350 (45.9) | 130 (44.1) | 220 (47.0) |
| Overweight | 299 (39.2) | 120 (40.7) | 179 (38.3) |
| Obese | 79 (10.4) | 33 (11.2) | 46 (9.8) |
| Marital status |  |  |  |
| Has a partner | 611 (80.1) | 213 (72.2) | 398 (85.0) |
| Not married | 29 (3.8) | 10 (3.4) | 19 (4.1) |
| Divorced or widowed | 123 (16.1) | 72 (24.4) | 51 (10.9) |
| Education |  |  |  |
| Illiterate | 285 (37.4) | 137 (46.4) | 148 (31.6) |
| Less than elementary school | 303 (39.7) | 117 (39.7) | 186 (39.7) |
| Junior high school or more | 175 (22.9) | 41 (13.9) | 134 (28.6) |
| Employment |  |  |  |
| Not currently employed | 189 (24.8) | 84 (28.5) | 105 (22.4) |
| Farming/fishing | 416 (54.5) | 169 (57.3) | 247 (52.8) |
| Self-employment | 31 (4.1) | 12 (4.1) | 19 (4.1) |
| Formal employee | 10 (1.3) | 2 (0.7) | 8 (1.7) |
| Part-time job with heavy physical activity | 43 (5.6) | 9 (3.1) | 34 (7.3) |
| Part-time job with low-moderate physical activity | 67 (8.8) | 16 (5.4) | 51 (10.9) |
| Others | 7 (0.9) | 3 (1.0) | 4 (0.9) |
| Household income |  |  |  |
| Low | 335 (43.9) | 156 (52.9) | 179 (38.3) |
| Middle | 324 (42.5) | 99 (33.6) | 225 (48.1) |
| High | 104 (13.6) | 40 (13.6) | 64 (13.7) |
| Alcohol consumption |  |  |  |
| Does not drink | 564 (73.9) | 232 (78.6) | 332 (70.9) |
| 1 or 2 days a week | 48 (6.3) | 20 (6.8) | 28 (6.0) |
| 3 to 6 days a week | 13 (1.7) | 4 (1.4) | 9 (1.9) |
| Every day | 71 (9.3) | 21 (7.1) | 50 (10.7) |
| Every day (heavy) | 67 (8.8) | 18 (6.1) | 49 (10.5) |
| Smoking |  |  |  |
| Never smoked | 605 (79.3) | 237 (80.3) | 368 (78.6) |
| Has stopped smoking | 53 (7.0) | 20 (6.8) | 33 (7.1) |
| Currently smoke | 105 (13.8) | 38 (12.9) | 67 (14.3) |
| Physical activity |  |  |  |
| Low | 241 (31.6) | 112 (38.0) | 129 (27.6) |
| Middle | 224 (29.4) | 79 (26.8) | 145 (31.0) |

*(Continued)*

**Table 1.** (Continued)

| | Total | Loneliness | |
|---|---|---|---|
| | (n = 763) | Yes (n = 295) | No (n = 468) |
| High | 298 (39.1) | 104 (35.3) | 194 (41.5) |
| Instrumental support | | | |
| None | 537 (70.4) | 231 (78.3) | 306 (65.4) |
| Receipt only | 27 (3.5) | 11 (3.7) | 16 (3.4) |
| Provision only | 13 (1.7) | 4 (1.4) | 9 (1.9) |
| Both receipt and provision | 186 (24.4) | 49 (16.6) | 137 (29.3) |
| Emotional support (Number of friends) | | | |
| None | 367 (48.1) | 162 (54.9) | 205 (43.8) |
| 1 to 5 | 256 (33.6) | 89 (30.2) | 167 (35.7) |
| 6 or more | 140 (18.4) | 44 (14.9) | 96 (20.5) |

BMI; body mass index. Mean [standard deviation] or n (%) are shown.

while 46.3% met the criteria for hypertension. The mean SBP and DBP was 140.9 and 82.6 mmHg among those who felt lonely, while it was 133.2 and 79.6 mmHg among those who did not feel lonely. As for social support variables, 70.4% neither received nor provided instrumental support while 24.4% received and provided it, and 48.1% answered they have no friends while 18.4% had 6 or more friends one can turn to for help (i.e., emotional support).

## Loneliness, social support, and hypertension

Poisson regression analysis revealed that loneliness was positively associated with hypertension (prevalence ratio [PR] 1.12, 95% confidence interval [CI], 0.99–1.26; Model 1 in Table 2). This

**Table 2. Results of Poisson regression model with robust variance estimator examining the association between loneliness, social support and hypertension among rural community dwellers in Fujian Province, China (n = 763).**

| | Model 1 | Model 2 | Model 3 |
|---|---|---|---|
| Loneliness (ref. Low) | 1.12 (0.99, 1.26) | 1.04 (0.89, 1.22) | 1.05 (0.91, 1.20) |
| Instrumental support (ref. None) | | | |
| Receipt only | | 1.08 (0.75, 1.57) | 1.07 (0.75, 1.53) |
| Provision only | | 1.07 (0.59, 1.95) | 1.09 (0.69, 1.73) |
| Both receipt and provision | | 0.99 (0.83, 1.19) | 1.14 (0.98, 1.33) |
| Emotional support (ref. Low) | | | |
| Middle | | 0.91 (0.80, 1.05) | 0.88 (0.75. 1.03) |
| High | | 0.83 (0.73, 0.95) | 0.72 (0.60, 0.87) |
| Loneliness x Instrumental support (ref. None) | | | |
| Receipt only | | 0.95 (0.58, 1.58) | |
| Provision only | | 1.08 (0.32, 3.57) | |
| Both receipt and provision | | 1.49 (1.20, 1.87) | |
| Loneliness x Emotional support (ref. Low) | | | |
| Middle | | | 1.09 (0.92, 1.28) |
| High | | | 1.44 (0.90, 2.33) |

Values are prevalence ratios and 95% confidence intervals. Covariates included age, age-squared, sex, body mass index category, marital status, educational attainment, employment status, household income, alcohol consumption, smoking and physical activity. A random effects model was used to account for multiple individuals in each community.

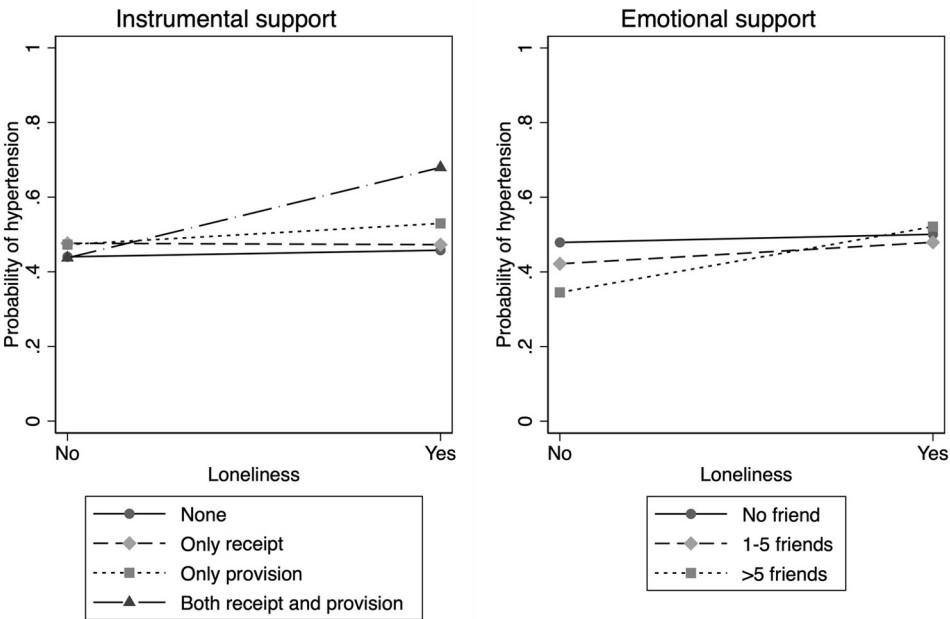

**Fig 1. Interaction of the relationship between loneliness and social support on hypertension.** Models are controlled for age, sex, body mass index, marital status, educational attainment, alcohol consumption, smoking, and physical activity. The y-axis represents predicted probability.

association was also observed when SBP (coefficient = 0.021, 95% CI, 0.001–0.043) or DBP (coefficient = 0.021, 95% CI, 0.001–0.040) were used as log-transformed continuous outcomes.

The analyses of statistical interaction (Models 2 to 4 in Table 2) showed that relation between loneliness and hypertension was more pronounced among those who reported *higher* social support compared to those who reported lower support. As shown in Fig 1, in the case of instrumental support, people who did not feel lonely had similar risks of hypertension regardless of the level of support. However, among those who reported feeling lonely, higher instrumental support (both receipt and provision) was associated with an *increased* prevalence of hypertension. That is, there was a direction of interaction that was *opposite* to the prediction of the buffering hypothesis. Among people who do not feel lonely, higher levels of emotional support from friends was weakly correlated with lower prevalence of hypertension. However, among individual who felt lonely, everyone converged to a similar risk of hypertension regardless of level of emotional support, i.e., there was no evidence in support of the buffering hypothesis. RERI for the interaction between loneliness and both receiving and providing instrumental support was 0.67 (95% CI 0.39–0.95, $p < 0.001$) in Model 2, while that for interaction between loneliness and emotional support was 0.17 (95% CI 0.01–0.34) for 1–5 friends, 0.32 (95% CI -0.14–0.78) for 6 or more friends, respectively. That is, the direction of results for additive interaction were consistent with those from the multiplicative interaction.

## Sensitivity analysis

When we excluded covariates from the analyses, the observed associations were statistically significant (i.e., loneliness was associated with hypertension, and there were significant interactions between loneliness and social supports on hypertension) while the magnitude of the estimates became larger; for example, the association between loneliness and hypertension was stronger (PR = 1.33, 95% CI 1.18–1.51 in Model 1). When we analyzed different cut-points for

loneliness (i.e., frequently/always vs. sometimes/rarely/never), the results were attenuated (PR = 1.06, 95% CI 0.89–1.26]) but the overall trends and patterns did not change across models. When we conducted the analyses stratified by age group, the significant association was only observed among those aged 65 years or older (PR = 1.26, 95% CI 1.06, 1.51, n = 269), while there was no significant association among younger people (PR = 0.81, 95%CI 0.44, 1.49 among those aged 18–44 years (n = 105); PR = 1.02, 95% CI 0.81–1.27 among those aged 45–64 years (n = 389)). The interactions between social support and loneliness were also significant among those aged 65 years or older for both instrumental support (loneliness x provision only: PR = 2.05, 95% CI 1.14, 3.71, loneliness x both receipt and provision: PR = 1.47, 95% CI 1.06, 2.06) and emotional support (loneliness x middle support: 1.59 95%CI 1.15, 2.19, loneliness x high support: PR = 1.31, 95% CI 0.99, 1.74). When we ran the analysis with community fixed effects rather than random effects, the effect size for the association between loneliness and hypertension became smaller but trended in the same direction (PR = 1.10, 95% CI 0.96, 1.26).

## Discussion

### Summary of the findings

In a cross-sectional sample of 763 rural community dwellers in Fujian Province, China, individuals who reported feeling lonely were more likely to have hypertension than those who did not feel lonely, especially among older people. In addition, we found significant interaction between loneliness and social support in relation to hypertension. More specifically, the positive association between loneliness and hypertension was more pronounced among those who reported higher social support compared to those who reported lower support.

### Loneliness and hypertension

Our finding in relation to the association between loneliness and hypertension is in line with previous studies. For example, a cross-sectional survey among 1,880 older Malaysians showed that lonely individuals had a higher likelihood of hypertension (6). Also, a longitudinal study among 229 participants in the U.S. found that loneliness predicted increased systolic blood pressure over a 4-year period [5]. One possible pathway linking loneliness and hypertension is the behavioral pathway. Hawkley and Cacioppo [3] have argued that loneliness is equivalent to feeling unsafe and those who feel lonely have higher sensitivity to social threat in the environment; lonely individuals see the social world as a more threatening place, expect more negative social interactions, and remember more negative social information. Social threat can cause diminished self-regulation, so that people with loneliness tend to have worse health behaviors such as more alcohol consumption and less physical activity [31, 32]. In our study sample, those who felt lonely reported lower physical activity (38.0% vs. 27.6%), while physical activity was not clearly associated with hypertension in this study. In addition, less heavy drinking was observed for those reporting loneliness (6.1% vs. 10.5% engaged in daily heavy drinking, defined as 50g or more pure alcohol)). These patterns contradict previous reports in Western settings where lonely people have been found to exhibit unhealthier habits. Given that the inclusion of health behavior variables did not weaken the association between loneliness and hypertension (not shown in Tables), we can at least conclude that in this study, lifestyle differences do not mediate the association between loneliness and hypertension. In rural China, every household typically cultivate their own rice field, and two-thirds of the study participants answered that they engaged in farm work on a daily basis. Hence people in our sample have relatively higher daily physical activity levels than is typical in urban settings, and lower physical activity compared to other people in the same community among the study participants

may not necessarily be linked with higher risk for hypertension. In our rural setting, drinking and smoking are also important tools for social interaction, so that the social context of these behaviors also differ from other settings in which loneliness has been studied. A previous study that used data from 29 districts of 3 cities in China found that high membership rate in social organizations was associated with higher prevalence of harmful drinking among both men and women. The authors concluded that the Chinese drinking culture may influence drinking behaviors [33].

Another possible pathway is a pathophysiological pathway. Loneliness can be conceptualized as a chronic stressor. Chronic stress (e.g., long-term activation of the hypothalamic-pituitary-adrenocortical (HPA) axis) has been shown to result in allostatic load [34], resulting in chronic overproduction of stress hormones (e.g., cortisol) which can lead to elevated blood pressure in the long term [5]. In this study sample, loneliness was significantly associated with stress measured by the Kessler Psychological Distress Scale (K6) [35] among those aged 70 years or younger, which was not investigated among older people since there was difficulty understanding Mandarin Chinese (older people in this area usually communicate in dialect) (mean [SD] 3.7 [3.8] vs. 6.7 [5.4]).

## Social support and hypertension

As for instrumental support, those with higher social support showed much higher prevalence of hypertension when they were lonely. One possible explanation is that in communities with a comparative absence of working-age adults, those who actively engage in social exchange might also be burdened with a bigger workload and experience greater stress, and thus more likely to establish hypertension. This is so-called 'dark side of social capital' [36, 37] which has been previously reported for example, in impoverished communities in the U.S. [38], rural Malawi [39] and rural China [33]. In our previous study which used the same dataset, those who participated more in wedding parties, funerals and social gatherings reported higher stress [22]. Chinese culture (along with other East Asian societies) maintains a tradition of gift exchange for lubricating social relationships with others. Gifts are exchanged between family and friends, but also between co-workers and business associates on significant occasions, such as the Chinese New Year or the mid-autumn festival. In rural areas, gift exchange within social networks functions as a form of informal insurance, creating an obligation on the part of the recipient to reciprocate a favor in the future [40]. For individuals who are feeling lonely, customs surrounding gift exchanges can be a form of social stress [41]. Although we hypothesized that emotional support would buffer the association between loneliness and hypertension, we rather found the opposite trend, viz., among individuals who felt lonely, they converged to a similar risk of hypertension regardless of level of emotional support. Having a lot of friends appeared to be associated with better health (i.e., low risk for hypertension), but this protection was not observed if they felt lonely. We used the number of close friends to whom people could turn to for help when they are in trouble as an indicator of expected emotional support between close friends since a systematic review have shown that a stress-buffering effect is most consistently found when support is measured as a perception that one's network is ready to provide aid and assistance if needed [42], while it is possible that it also reflects a trait personality factor (e.g., optimism) and bigger social network size. Moreover, as mentioned in the above discussion, the social norms and social context of health behaviors in rural China can be different from those typically found in Western settings. A study in China has shown that those who participate in social activity (e.g., interact with friends, playing mahjong or cards) were less likely to establish hypertension after two years [43]. We found that those who are lonely had low level of social capital (i.e., trust of others, community attachment,

and reciprocity among community members) (the unadjusted correlation = −0.11 for trust, −0.17 for attachment, and −0.22 for reciprocity; S1 File), and those reporting higher perceptions of social capital reported more friends (chi-squared test: $p < 0.001$ for trust, 0.013 for attachment, 0.003 for reciprocity), which may indicate that emotional benefits such as an increase in sense of belonging or purpose through having friends may be more closely linked to hypertension. This may also relate to the finding that there was a main effect of emotional social support on risk of hypertension, which was in accordance with a U.S. study showed that those with emotional support from friends were less likely to have uncontrolled and undiagnosed hypertension [44].

## Limitations

This study has several limitations. First, it is not possible to infer causality from cross-sectional design. For example, it is possible that lonely people seek out the emotional support of friends at the same time as developing hypertension (simultaneity), or poor health may lead to increased loneliness and more attempts to seek social support (reverse causality). Previous literature has documented that the amount of received support may reflect poor health status especially in cross-sectional settings [45, 46]. This has been called the support mobilization hypothesis [47], which posits that received support is an indicator of poor health. However, in this study, those who reported feeling lonely were less likely to drink alcohol or to smoke, even though they reported fewer supports. Those who received more instrumental support tended to drink less and smoke, while those reporting more friends tended to be younger, had higher education/income, smoked more, and were more physically active. These patterns tend to suggest that healthier people receive more support (see S1 Appendix). We also note that loneliness could have been the product of social support, i.e., those receiving social support tended to feel less lonely. While we assumed temporal ordering from loneliness → social support, the association between loneliness and social support is likely to be bidirectional. Hence, exchange of social support can be simultaneously a mediator and confounder of the association between loneliness and hypertension. The bidirectionality cannot be teased out in cross-sectional data. It would be also challenging to tease out in longitudinal data, without multiple waves of data capturing changes in loneliness and social support. Second, the participants might not have fully represented adults living in rural communities in Fujian as the survey was conducted in only one city in Fujian and we did not use a random sampling procedure. Especially, it is possible that younger, healthier people left the areas to seek migratory work (healthy migrant hypothesis) and were not included in our survey. Third, the measurement of loneliness relied on one question, although this was because of the relatively low educational background of the participants (37% were illiterate). Fourth, we did not ask receipt and provision of emotional support separately. There are also other aspects of social support to be evaluated (i.e., informational support, appraisal support) and *quality* of social support was not assessed. Although we tried to address this issue by defining close friendships as those to whom people could turn to for help when they are in trouble, we did not inquire about the perceived quality of support received. Furthermore, our social support measures have not been validated against existing instruments. Fifth, some important variables were not available which enable us to better interpret the findings; for example, sleep quality [3], poor dietary habit especially salt intake and depression, which is distinct from loneliness [48] can be important confounders between loneliness and hypertension. Sixth, the measurement of blood pressure was conducted using an automated oscillometric monitor. Usage of a random-zero sphygmomanometer would have been ideal, but we did not use it due to time and resource restriction. Lastly, the exclusion of those with missing values might have biased the observed associations toward the null, e.g.,

those with missing values might have had health-related problems which prevented them from engaging in social interactions.

## Conclusions

This study showed that loneliness was positively associated with hypertension in rural Fujian communities in China, but the association was not buffered by social support. Overall, feeling lonely in spite of being surrounded by supportive alters in the network was most strongly linked with increased risk for hypertension. Due to the rapid aging of the population and the continued outflow of younger generations to urban centers, it is anticipated that the prevalence of loneliness will rise in rural communities in China [49], and future studies are warranted to address this issue.

## Supporting information

**S1 File. Measurement of social capital.**
(DOCX)

**S1 Appendix. Characteristics of social support receiver/provider (n = 763).**
(DOCX)

## Author Contributions

**Conceptualization:** Yosuke Inoue.

**Data curation:** Aki Yazawa, Yosuke Inoue.

**Formal analysis:** Aki Yazawa, Ichiro Kawachi.

**Funding acquisition:** Aki Yazawa.

**Investigation:** Aki Yazawa, Yosuke Inoue, Raoping Tu.

**Methodology:** Aki Yazawa, Chiho Watanabe.

**Project administration:** Aki Yazawa, Yosuke Inoue, Chiho Watanabe, Raoping Tu.

**Resources:** Taro Yamamoto, Chiho Watanabe, Ichiro Kawachi.

**Supervision:** Yosuke Inoue, Taro Yamamoto, Chiho Watanabe, Ichiro Kawachi.

**Writing – original draft:** Aki Yazawa, Yosuke Inoue.

**Writing – review & editing:** Taro Yamamoto, Chiho Watanabe, Raoping Tu, Ichiro Kawachi.

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
