## [Decision Letter · Decision Letter 0]

13 Oct 2021

PONE-D-21-28413Can social support buffer the association between loneliness and hypertension? A cross-sectional study in rural ChinaPLOS ONE

Dear Dr. Yazawa,

Thank you for submitting your manuscript to PLOS ONE. After careful consideration, we feel that it has merit but does not fully meet PLOS ONE’s publication criteria as it currently stands. Therefore, we invite you to submit a revised version of the manuscript that addresses the points raised during the review process.

We look forward to receiving your revised manuscript.

Kind regards,

Akihiro Nishi, M.D., Dr.P.H.

Academic Editor

PLOS ONE

Journal Requirements:

[AY is financially supported by Japan Society for the Promotion of Science as a JSPS Overseas Research Fellow.]

 [This study was financially supported by the JSPS KAKENHI from the Japan Society for the Promotion of Science (JSPS KAKENHI Grant Number JP13J06172; recipient: AY). The funders had no role in study design, data collection and analysis, decision to publish, or preparation of the manuscript.]

Additional Editor Comments:

Please follow the reviewers comments. I am looking forward to seeing the revision.

Reviewers' comments:

Reviewer's Responses to Questions

**Comments to the Author**

1. Is the manuscript technically sound, and do the data support the conclusions?

Reviewer #1: Partly

Reviewer #2: Yes

2. Has the statistical analysis been performed appropriately and rigorously? 

Reviewer #1: Yes

Reviewer #2: Yes

3. Have the authors made all data underlying the findings in their manuscript fully available?

Reviewer #1: No

Reviewer #2: Yes

4. Is the manuscript presented in an intelligible fashion and written in standard English?

Reviewer #1: Yes

Reviewer #2: Yes

5. Review Comments to the Author

Reviewer #1: PONE-D-21-28413_reviewer

General comments

Thank you for giving me the opportunity to review this research about the association of the sense of loneliness and blood pressure, which I am sure is an important topic in the aging society. In this cross-sectional study, people with the sense of loneliness were more likely to have high blood pressure, but the statistical association was pronounced once social relationships were adjusted. I have some issues the authors should consider.

Comments

Introduction

1. Line 77–: I understand that the aim of this study was to evaluate 1) the association of the sense of loneliness and high blood pressure in rural China and 2) the role of social relationships in the association. Thus, it seems to me that “the specific mechanisms linking each type of social support and health outcomes remain unclear” is not relevant.

2. Line 82 & 83: I am confused with your usage of “positive” and “negative” in the last sentence of Introduction; I would replace “negative” with “positive.”

Materials and methods

1. Line 89–: You mentioned “the basis of average population size and level of economic development.” I think you should be more specific and explain the criteria you adopted.

2. Line 95–: I am wondering how the difference of the excluded samples and the included ones could impact on the estimates. Could you mention the possible impact in Discussion?

3. Line 116–: I am wondering why only “receiving emotional support” was evaluated and “providing” emotional support was not. Furthermore, how could you claim that the stated number of friends represented just the “reception” of emotional support, instead of reciprocal friendship? I am also concerned about the validity of the three measures of social support.

4. Line 127–: I think income and employment status were important confounders when considering the association of the sense of loneliness and health. Could you include these variables in this study? If you do not have the information, in what direction should the omitted-variable bias be?

5. Line 138–: How about introducing community fixed-effects to account for time-invariant community specific characteristics?

6. Line 148–: I think SBP and DBP were used without making it clear what they stood for. Could you be specific?

Results

1. Table 1: I am wondering why Table 1 was presented according to the age group. I would replace this table by the table in Appendix. Furthermore, what “[]” and “()” indicated should be clear in this table.

2. Line 178–: Why were the mean blood pressure presented here? This information should rather be placed in “Characteristics of study participants.”

3. Line 182: “Fig 1” should be spelled out as “Figure 1.”

4. Line 184: I am not sure whether the expression “lonely people” is appropriate; the sense of loneliness in this study was just a measurement, and some readers might think this expression could induce a form of discrimination.

5. Line 185–: Importantly, we cannot interpret a mere statistical interaction as a biological interaction (a mechanism). How about calculating superadditivity, referring to Chapter 5 in Modern Epidemiology 3rd Edition?

6. Table 2: I would present the estimates and confidence intervals only for the independent variable and interaction terms. You should not present irrelevant rows, all the more so since you did not try to find risk factors.

7. Figure 1: The subtitles of the right two graphs should be different. I would also add confidence intervals for each point.

8. Line 198–: I am wondering what the expression “the observed associations were all unchanged and more obvious” means. Could you be more specific?

Discussion

1. Line 205–: I think you should mention that this study was cross-sectional and the direction of the statistical interaction in the summary paragraph. Furthermore, I would replace the expression “individuals who reported feeling lonely showed a higher prevalence” because prevalence is defined for population, not for a person.

2. Line 215: I do not think the odds ratio of the previous study is necessary unless the figure has a meaning.

3. Line 219: I think you should make it clear what the social threat is.

4. Line 222–: You cannot judge whether people who felt lonely were more likely to be engaged in some healthy behaviors because those associations were subject to confounding. There is a concern for multiple-comparisons, too.

5. Line 222–: The sentence “In our rural setting, drinking and smoking” seems to me subjective. Could you present some evidence to this claim?

6. Line 235–: I am wondering whether you could check the variable for stress; otherwise, it is not clear what this paragraph was aimed for.

7. Line 249–: Could you please explain what the gift-giving culture? Do you have some information on this feature in your sample?

8. Line 250–: This paragraph should be placed on the Limitation subsection. Furthermore, the expression “inverse association” does not make sense; an association can be bidirectional contrary to a causal relationship.

9. Line 261– & Limitation: It is important to keep in mind that this is a cross-sectional study, although a longitudinal study does not solve all the issues; how could you tell social relationships were not a confounder but a mediator? If they were just a confounder, the results were not incompatible with the buffering hypothesis at all. Rather, the results without the adjustment for social relationships might be confounded. How could you justify to claim the association was “pronounced” instead of “less biased” under the possibility of confounding?

Others

1. I have the impression that this manuscript needs a revision by a native English proofreader, although I do not feel qualified to judge English as it is not my mother language. Hence, I might fail to get the gist of this manuscript, especially for Discussion.

2. Conclusion should be concise; new Discussion cannot be included there.

I hope you can make use of the primary data in a clear and transparent way and contribute to the literature on social relationships and health.

Reviewer #2: The authors examined associations of interaction between loneliness and social support with prevalent hypertension and found paradoxical associations: individuals with both loneliness and high social support were more likely to have hypertension than those with either loneliness or low social support. Those findings may be a fact but look like an artifact. I have several concerns below.

Comments

1. According to table 1, participants aged 65 years or older look substantially different from those aged 64 years or younger. Since the late '90s, Chinese economic growth became great, and intranational migration from rural areas to industrial or urban areas drastically increased as the authors mentioned in the introduction. This historical background may cause selection bias and result in paradoxical associations. To avoid such historical selection bias, authors should show the results stratified by age as well.

2. Alcohol intake is assessed only as frequency, not as the amount. Also, the information about quitting alcohol intake was not available. The dose-response association between alcohol intake and incident hypertension has been well-established. However, such an association for prevalent hypertension did not find in the present study. This discrepancy may happen because quitting or reducing alcohol intake was more likely to occur among hypertensive patients and because alcohol intake was misclassified due to the lack of data on the amount of alcohol intake. Furthermore, the present paradoxical findings may be explained by the binge drinking, not assessed in the present study, among individuals with loneliness and high social support.

3. In table 3, women were more likely to have hypertension than men. Is this true? In general, men are more likely to have hypertension than women. The authors need to confirm the reference category of sex in table 3. If no error, there may be selection bias between men and women.

4. In table 3, age is better adjusted for as fifths or other categories than the continuous value because the association between age and hypertension can be non-linear rather than linear. In addition, for BMI, the reference category should have a narrower range such as 18.5-22.9 kg/m2. The overweight criterion for Asians is 23.0 kg/m2.

6. PLOS authors have the option to publish the peer review history of their article (what does this mean?). If published, this will include your full peer review and any attached files.

Reviewer #1: No

Reviewer #2: No

---

## [Author Response · Author response to Decision Letter 0]

6 Jan 2022

We have attached the response letter as a Word file.

---

## [Decision Letter · Decision Letter 1]

3 Feb 2022

Can social support buffer the association between loneliness and hypertension? A cross-sectional study in rural China

PONE-D-21-28413R1

Dear Dr. Yazawa,

We’re pleased to inform you that your manuscript has been judged scientifically suitable for publication and will be formally accepted for publication once it meets all outstanding technical requirements.

Kind regards,

Akihiro Nishi, M.D., Dr.P.H.

Academic Editor

PLOS ONE

Additional Editor Comments (optional):

I am happy to accept the manuscript. Please do the final edit (including the one suggestion from the reviewer 2) during the publication process.

Reviewers' comments:

Reviewer's Responses to Questions

**Comments to the Author**

1. If the authors have adequately addressed your comments raised in a previous round of review and you feel that this manuscript is now acceptable for publication, you may indicate that here to bypass the “Comments to the Author” section, enter your conflict of interest statement in the “Confidential to Editor” section, and submit your "Accept" recommendation.

Reviewer #1: All comments have been addressed

Reviewer #2: All comments have been addressed

2. Is the manuscript technically sound, and do the data support the conclusions?

Reviewer #1: Yes

Reviewer #2: Yes

3. Has the statistical analysis been performed appropriately and rigorously? 

Reviewer #1: Yes

Reviewer #2: Yes

4. Have the authors made all data underlying the findings in their manuscript fully available?

Reviewer #1: No

Reviewer #2: Yes

5. Is the manuscript presented in an intelligible fashion and written in standard English?

Reviewer #1: Yes

Reviewer #2: Yes

6. Review Comments to the Author

Reviewer #1: For the last part of Discussion, please consider removing "toward the null" from "Lastly, the exclusion of those with missing values might have biased the observed associations toward the null, [...]"

Reviewer #2: Thank you for carefully addressing my concerns. I do not have any further comments on this manuscript.

7. PLOS authors have the option to publish the peer review history of their article (what does this mean?). If published, this will include your full peer review and any attached files.

Reviewer #1: No

Reviewer #2: **Yes: **Isao Muraki

---

## [Editor Report · Acceptance letter]

8 Feb 2022

PONE-D-21-28413R1 

Can social support buffer the association between loneliness and hypertension? A cross-sectional study in rural China 

Dear Dr. Yazawa:

I'm pleased to inform you that your manuscript has been deemed suitable for publication in PLOS ONE. Congratulations! Your manuscript is now with our production department. 

Kind regards, 

on behalf of

Dr. Akihiro Nishi 

Academic Editor

PLOS ONE